# Comparative Histopathological and Morphometric Analysis of Lung Tissues in Stillborn Cubs of South China Tiger and Amur Tiger

**DOI:** 10.3390/biology14070833

**Published:** 2025-07-08

**Authors:** Le Zhang, Jincheng Yang, Fengping He, Yaohua Yuan, Zhaoyang Liu, Guangyao Geng, Kaixiong Lin, Qunxiu Liu, Dan Liu, Tianlong Liu, Yanchun Xu

**Affiliations:** 1College of Wildlife and Protected Area, Northeast Forestry University, Harbin 150040, China; zhangl_0811@163.com (L.Z.); maturecheng@163.com (J.Y.); 2College of Veterinary Medicine, Yunnan Agricultural University, Kunming 650201, China; hefengping@outlool.com; 3Shanghai Zoological Park, Shanghai 200335, China; knhuang@163.com (Y.Y.); gengshzoo@163.com (G.G.); 4Luoyang Zoo in Wangcheng Park, Luoyang 471000, China; lyswcgydwy@126.com; 5Fujian Meihuashan Institute of South China Tiger Breeding, Longyan 364201, China; mhslkx@163.com; 6School of Health & Social Care, Shanghai Urban Construction Vocational College, Shanghai 201415, China; liuqunxiu@126.com; 7Heilongjiang Siberian Tiger Park, Harbin 150040, China; liudan_1964@sina.com; 8College of Veterinary Medicine, China Agricultural University, Beijing 100089, China; 9National Forestry and Grassland Administration Research Center of Engineering Technology for Wildlife Conservation and Utilization, Harbin 150040, China

**Keywords:** South China Tiger, Amur Tiger, lung, hypoxia

## Abstract

The South China Tiger (*Panthera tigris amoyensis*) is classified as a nationally protected species under the highest conservation category. As this tiger is functionally extinct in the wild, only captive populations remain. Prolonged captivity has led to inbreeding, resulting in severe challenges from inbreeding depression. The reduced reproductive fitness in adults and the mortality rate of young tigers will seriously restrict the sustainable development of the population. To elucidate the underlying causes of elevated cub mortality, this study conducted comparative histopathological and morphometric analyses of lung tissues from stillborn South China Tiger and Amur Tiger (*P. t. altaica*) cubs. The findings provide crucial scientific insights into the survival challenges confronting endangered species and offer valuable theoretical support for genetic management strategies for the South China Tiger.

## 1. Introduction

The South China Tiger (*Panthera tigris amoyensis*, SCT), an endemic Chinese subspecies [1], has been listed as extinct in the wild (EW) by the International Union for Conservation of Nature [2]. Fortunately, a captive population consists of more than 200 individuals, descended from six founders (four females and two males) captured in the wild between the late 1950s and the 1970s [3]. Currently housed in multiple zoos across China, this population has experienced inbreeding since at least 1972, with its prevalence increasing over time [3,4]. The average inbreeding coefficient ranged from 0.2586 in 1999 to 0.3550 in 2019, while genetic diversity declined to 65.23% [5,6]. The increasingly severe inbreeding depression affects reproductive performance and juvenile survival [4]. An analysis of neonatal mortality across five primary SCTs between 2009 and 2018 indicated that 52.17% of cub deaths were attributable to premature birth (17.39%) and stillbirth (34.78%) [5]. Notably, the rates of stillbirth (that is, any animal that was born dead or died on the day of its birth) were significantly higher than those reported for Siberian (*P. t. altaica*) and Sumatran Tigers (*P. t. sumatrae*), which stood at 11.11% [7]. In a recent study examining the survival rates of captive Bengal Tigers (*P. t. tigris*) at Lahore Zoo, researchers identified stillbirth (with a rate of 37%) and rickets as significant issues related to inbreeding among these tigers [8]. Understanding the pathogenesis of stillbirth is crucial for formulating strategies to save tiger subspecies from extinction.

Etiologically, stillbirth can be attributed to both infectious and noninfectious causes. Infectious causes include maternal infections with spirochetes, protozoa, viruses, and bacteria [9]. Noninfectious causes are genetic, maternal, fetal, and environmental factors [10]. Our investigation revealed that most stillborn SCT cases presented normal developmental parameters, with delivery occurring at the expected gestational dates. The tigresses were rarely recorded to encounter infectious or other diseases during pregnancy. Thus, it could be conjectured that all such fetuses developed normally and were alive until the onset of partum when they encountered fatal factors.

Evidence shows that obligatory or transient asphyxia experienced by newborns during birth is a common phenomenon across all mammalian species [11,12,13]. Asphyxia can trigger a cascade of cellular biochemical events, resulting in temporary alterations in cellular function and/or cell death. These changes manifest as organ impairment, dysfunction, and failure, ultimately contributing to prenatal and postnatal mortality [14]. For this study, we collected six stillbirth cub specimens from SCTs and Amur Tigers (*P. t. altaica*, AT), where stillbirth is defined as any animal that was born dead or died on the day of their birth [7]. Subsequently, lung histopathological examinations were performed to evaluate the potential contribution of asphyxiation to neonatal mortality.

## 2. Materials and Methods

### 2.1. Tissue Collection

A total of six neonatal tigers were collected from zoos, and the detailed information regarding these samples is presented in Table 1. Tissue samples were collected from individuals who died of natural causes and were preserved in zoo veterinary hospitals. All samples were conducted with due permission in compliance with the local license. No samples of wild tiger cubs were included in this study. The experimental designs were approved by the Scientific Ethics Committee of Northeast Forestry University (No. 20191203, Harbin, China; 9 December 2019).

### 2.2. Appearance Examination and Necropsy

Appearance examinations were performed for all six individuals immediately after the samples were obtained, including examination of the integrity and foreign matter of the eyes, oral cavity, nasal cavity, anus, external reproductive organs, and hair coat. Body weight and length were measured using an electronic balance (precision: 1 g) and a calibrated ruler (precision: 0.1 cm), respectively. The internal organs, including the heart, liver, kidneys, lungs, and bronchi, were systematically examined for color, morphology, presence of lesions, hemorrhagic spots, and signs of necrosis.

### 2.3. Histological Examination and Special Stains

Ten tissue samples, each measuring approximately 5 mm × 5 mm × 5 mm, were excised from the lower right lung lobes of six individual tigers. All the tissue samples were subsequently fixed in a mixture of 2.5% glutaraldehyde and polyformaldehyde fixative buffer at 4 °C for 48 h. Each tissue sample was dehydrated using an ethanol–water mixture at concentrations of 70%, 80%, 90%, and 95% for 1 h during each phase. A 5 μm slide with a cross-section in the transverse plane of the lung was prepared according to routine procedures. The slides were stained with hematoxylin and eosin (H&E) and examined under a microscope. Additionally, special staining was performed using Prussian blue and Alcian blue (Solarbio, Beijing), strictly following the manufacturer’s instructions. For further analysis, immunohistochemical staining was conducted using high-molecular-weight keratin (CK5/6) to assess specific tissue markers. The morphology of the alveoli and their ectasia were observed under a Scanner M8 (PreciPoint GmbH & Co. KG, Garching bei München, Germany).

### 2.4. Morphometric Analysis of the Lung

Randomly selected nonoverlapping ×200 magnification fields of view from each slide were imaged for subsequent statistical analysis. The spatial calibration and semiquantitative image analysis were performed with Image-Pro Plus (v7.0) [15]. The irregular shape selection tool for selecting alveolar spaces (*P(ASP)*) enables measurements of the alveolar area and the mean alveolar linear intercept (*L_m_*). Morphometric analysis of the mean thickness of the alveolar septa (*Tas*) was performed by superimposing 50 randomized linear measurements perpendicular to the septal walls via digital image analysis software. The average of these measurements was subsequently calculated. Morphological statistics and visualization of the results were performed via R (v4.2.2) [16].

## 3. Results

### 3.1. Visual Examination and Necropsy Report

SCT_stb1 (highly inbred individual): This full-term born dead fetus showed normal development, weighing 1219 g with a body length of 27.3 cm. Gross examination revealed no macroscopic abnormalities, with no evidence of lesions, hemorrhages, or necrosis in any of the examined organs (esophagus, gastrointestinal tract, liver, pancreas, or kidneys). Although the tracheobronchial structures appeared normal, the lungs exhibited dark red discoloration, compressive atelectasis, and prominent petechial hemorrhages.

SCT_stb2 (highly inbred individual): Surveillance footage confirmed that the neonate expired within 15 min postdelivery. The newborn measured 1250 g and 28.2 cm in length, with no observable abnormalities in external organ development. During necropsy, hemorrhage was found in the abdominal cavity, the lungs were bright red, and the liver surface showed scattered millet-like white spots. There were no apparent changes in the other organs. Among the remaining siblings, one exhibited extreme weakness, whereas the other demonstrated incomplete skull closure.

SCT_alv (highly inbred individual): This cub, a littermate of SCT-stb1, was delivered alive but displayed severe neonatal weakness with markedly reduced mobility compared to healthy counterparts. The neonate failed to initiate suckling and succumbed at 5 h postpartum, with recorded morphometric measurements of 1455 g body weight and 32.2 cm body length. All the appearance examinations and necropsies revealed neither abnormalities in the organs nor signs of disease. The lungs appeared dark red and spongy, contrasting with those of its littermate SCT-stb1, and petechial hemorrhages were also observed.

AT_utd (non-inbred individual): This fetus was obtained from the uterus after the mother died in an accident approximately 5 days before the expected date of confinement. It was fully developed, with a body mass of 1189 g and a body length of 26.5 cm. The autopsy results revealed that all the organs were structurally intact, with no apparent abnormalities detected.

AT_stb1 (non-inbred individual): This cub was the terminal offspring in a litter of triplets and succumbed immediately postpartum. It had a significantly smaller body size relative to littermates, with recorded measurements of 1146 g in body weight and 24.6 cm in body length. A comprehensive external examination and necropsy revealed no apparent abnormalities or pathological signs in any organ. The lungs appeared dark red, and no petechial hemorrhages were detected.

AT_stb2 (non-inbred individual): This neonate was the firstborn of a litter of five tiger cubs, with a body mass of 1242 g, and exhibited normal physical development. It was weak after the birth and showed no signs of life in the surveillance footage a few minutes later. The fur was clean, and the lungs appeared light red (nearly white) and were larger than those of a South China Tiger (SCT_stb2). The lung tissue was observed floating in the fixative. The liver was dark red with a blackish-red anterior lobe, and the other organs were developing within normal limits.

### 3.2. Histopathological and Morphometric Changes in the Lung Tissue

H&E staining revealed that all six samples exhibited partial expansion of alveolar spaces (Figure 1). This supports the possibility of having breathed after birth or having performed deep inspiratory movements before birth, which were sufficient to partially expand the lungs. In each animal, the alveolar septa appeared relatively thick and hypercellular compared to the normally expanded alveoli. The histological observations for all samples were as follows:

SCT_stb1 (Appendix A and Figure 1A): Extensive hemorrhage in the alveolar cavities, with purplish exudate observed in some areas. Interstitial edema is pronounced and widening, accompanied by pulmonary edema. The interstitial vessels were dilated and congested, and there was abundant brown granular material within the alveolar epithelial cells (Figure 2A). Simultaneous identification through Prussian blue and Alcian blue staining indicated that these brown granules were composed of iron-containing hemosiderin deposits (Figure 2B,C) as well as meconium constituents (Appendix A).

SCT_stb2 (Appendix A and Figure 1B): The lung alveoli exhibit fusion and expansion, along with the inflammatory cell infiltration observed in the pulmonary interstitium. Additionally, it is accompanied by punctate pulmonary bruising.

SCT_alv (Appendix A and Figure 1C): Lung tissue exhibited significant hemorrhagic foci characterized by alveolar septal thickening, interstitial edema, and diffuse lymphocytic infiltration. The alveolar lumens were filled with eosinophilic proteinaceous exudate, and meconium components, as well as keratinized debris, were observed in the alveolar and bronchiolar lumens (Appendix A). Additionally, large irregular polygonal or rectangular deposits of eosinophilic homogeneous crystalline material were present in some alveolar interstitial spaces (Figure 3). The airway walls demonstrated compensatory dilatation, and the lamina propria exhibited edema with inflammatory cell infiltration.

AT_utd (Appendix A and Figure 1D): The alveolar cavities exhibited amniotic fluid obstruction with filamentous reticulated proteinaceous exudates are accompanied by subpleural and concurrent tracheal and vascular edema. Additionally, lymphocytic infiltration was observed in the alveolar septa, along with the thickening of the interstitial vasculature walls and a lax, purplish-red exudate within the tissue structure. Furthermore, the mucosa of the fine bronchi was found to be detached.

AT_stb1 (Appendix A and Figure 1E): The dilatation of the alveoli was not evident, but marked atrophy was observed. Additionally, peritracheal edema and widening of the interstitium were noted. A small quantity of exudate was detected within the alveolar lumen, accompanied by visibly desquamated epithelial cells and mucosal separation in the trachea.

AT_stb2 (Appendix A and Figure 1F): A prominent alveolar fusion expansion phenomenon is observed, accompanied by changes in emphysema. Furthermore, infiltration of inflammatory cells within the pulmonary interstitium was noted.

### 3.3. Morphometric Parameters

According to the measurement statistics, variations in *P(ASP)*, *L_m_*, and *Tas* were observed in samples from deceased tiger cubs across each group. Notably, the alveolar septal thickness was comparable between the Amur tiger (AT_utd) that experienced intrauterine death and the South China tiger (SCT_alv) that died 5 h postpartum. However, among neonates that succumbed immediately after birth, the South China tigers (SCT_stb1 and SCT_stb2) exhibited significantly greater alveolar septal thickness, approximately two-fold higher than their Amur tiger counterparts (AT_stb1 and AT_stb2). The results are presented in Table 2.

Based on our observations, the degree of alveolar expansion was lowest in SCT_stb1 among SCTs and in AT_utd among ATs (Figure 4A,B). Overall, there was a positive correlation between the expansion of the alveolar lumen and the linear intercept, and the trend was supported by our statistical results. The statistical analyses revealed significant differences in the alveolar septal thickness between the SCT_stb2 of neonatally deceased SCTs and the AT_stb2 of ATs (*p* = 5.26 × 10^−14^, Figure 4C). These variations were notable compared with the statistics concerning alveolar luminal dilatation and the linear intercept. Furthermore, data from SCTs and ATs at equivalent stages (stages of labor and birth) were integrated. The results revealed discrepancies in all three morphometric parameters between the two tiger subspecies, with a particularly pronounced distinction observed in the alveolar septal thickness (Figure 4D–F).

## 4. Discussion

### 4.1. Evidence for Intrapartum Hypoxia

Intrapartum hypoxia is a significant contributor to perinatal mortality in both humans and animals [17,18]. It often leads to systemic fetal damage, such as short-term hypoxic–ischemic encephalopathy, long-term cerebral palsy, and abnormal neurodevelopmental outcomes [19], as well as decompensation of basic cellular functions [20] and lung injury [21]. Considering the complete development of the fetus and on-time delivery, we expected intrapartum hypoxia to be the cause of the stillbirth of the SCT.

First, all specimens revealed partial expansion of the alveolar spaces (Figure 1A–F), consistent with the postnatal respiratory process that facilitates partial lung expansion. The fetus, which develops within a fluid-filled amniotic sac, relies on placental gas exchange rather than pulmonary function for respiration [22]. AT-utd experience intrauterine demise following maternal trauma from an accident. This event likely resulted in severe hypoxia as placental gas exchange ceased, potentially triggering deep inspiratory movements [23]. Similar occurrences have been documented in other cases (e.g., SCT-stb1 and SCT-stb2), supporting the hypothesis that interruption of placental gas exchange during labor can lead to fetal hypoxia and initiate respiratory movements [24,25,26].

Second, histological examination revealed the presence of foreign material within the alveolar spaces and respiratory bronchioles, identified as keratin fragments and meconium components (Figure 3, Appendix A). This finding is of significant importance. The identification of intra-alveolar meconium reinforces the concept of pathological respiratory movements arising from intrauterine distress [27,28], which may indicate a hypoxic stress response. Furthermore, inflammatory cell infiltration was observed in several tissue samples (e.g., Appendix A). Existing studies have established that the inhalation of meconium and amniotic fluid can result in neonatal aspiration syndrome, leading to subsequent chemical inflammatory responses [29,30].

### 4.2. Causes of Intrapartum Hypoxia

Histopathological analysis demonstrated that South China tigers exhibited significantly more severe partial congenital pulmonary hypoexpansion than Amur tigers (Figure 4D). The literature indicates this pathological manifestation represents a common neonatal respiratory disorder, leading to varying extents of pulmonary functional impairment [31]. The etiology of this condition is attributed to the interplay of multiple factors, including congenital developmental abnormalities [32], perinatal factors [33], and postnatal factors [34], among others. Notably, under captive conditions, the neonatal mortality rate of Amur and Sumatran tigers remains relatively low at 11% [7]. In contrast, the perinatal mortality rate of South China tiger cubs is strikingly high at 50% [5]. This significant inter-population disparity suggests that, after controlling for anthropogenic factors like husbandry management, the high mortality rate in South China tiger cubs is likely due to their distinct biological traits.

Studies have shown that the effects of inbreeding systematically influence multiple phenotypic traits, including birth length [35], forced expiratory lung volume in 1 s [36], and cognitive behavior [37], among other traits. Alarmingly, the captive South China tiger population is experiencing a progressive loss of genetic diversity [3,4,6]. Comprehensive genomic analyses further reveal that captive South China tigers show substantially reduced heterozygosity, lower single-nucleotide polymorphism (SNP) density, and diminished nucleotide diversity compared to Amur tigers [38,39,40]. Additionally, population genomic studies have identified that large segments of runs of homozygosity (ROH) and identical by descent (IBD) are prevalent in the genomes of captive SCT populations [38,39]. In contrast, this phenomenon is less pronounced in captive AT populations [41]. Importantly, veterinarians from multiple zoos nationwide have reported that pneumonia-related mortality in SCT cubs under 3 months was significantly higher than that in AT cubs of the same age under comparable rearing conditions. Combined with the statistical results of the SCT-stb1, SCT-stb2, and SCT-alv measurements (Figure 4C,F), the alveolar septa in the lung tissue of the SCT samples were considerably thicker than those in the AT cubs (AT_utd represents the intrauterine stage, while AT_stb1 and AT_stb2 represent the delivery stage) in all cases. We hypothesize that this structural difference may be directly linked to functional deficits in the respiratory system.

Additionally, the whole-genome sequencing has identified deleterious mutations in key developmental genes (e.g., *RCOR1*, *DENND1A*, *Myh10*, *PAPPA2*, etc.) within the SCT population [39]. Particularly noteworthy is the *Myh10* missense mutation (c.T1373G; p.L458R), which experimental studies confirm can induce pathological lung tissue thickening in mutant pups, leading to weak births and mortality [42]. Furthermore, gene knockout animal model studies have demonstrated the pleiotropic effects of inbreeding across multiple gene systems. For instance, deficiency in the vitamin D receptor (*VDR*) gene significantly impairs pulmonary function in mice [43], while homozygous knockout of the *BMPER* gene induces developmental abnormalities in both pulmonary and skeletal systems, ultimately leading to neonatal lethality [44,45].

Based on this, we hypothesize that prolonged inbreeding within the SCT population has reduced genetic diversity and increased the likelihood of expressing recessive deleterious mutations. At the same time, it may directly impair lung development and immune function genes, thereby contributing to the high mortality rates observed in offspring. Future comprehensive mechanistic studies are essential to address two critical questions: (1) whether the genetic diversity of lung development-related genes (e.g., *RCAN1* [46], *ACTN3* [47], *TGFβ* [48], *IGF-1R* [49], etc.) is affected by ROH regions; and (2) whether functional mutations in these genes directly impair pulmonary function, consequently leading to high cub mortality rates.

### 4.3. Conservation Implications

The wild SCT population is nearly extinct [2], and all captive individuals suffer from inbreeding [6,50,51]. Research has shown that inbreeding leads to widespread sharing of ancestral alleles within the SCT population, causing harmful alleles to become more prevalent through homozygous combinations [38,39]. Consequently, the high mortality rate is likely to persist as a normal feature of the population. Although targeted interventions during labor have effectively reduced perinatal and neonatal mortality [52], implementing such interventions poses significant challenges due to the unique characteristics of tigers.

In this study, we systematically elucidated the causes of mortality in South China tiger cubs for the first time and hypothesized the underlying mechanisms leading to increased mortality. Based on literature evidence [42,43,44,45,47,48], we believe that reduced genetic diversity and the accumulation of deleterious mutations in genes associated with lung tissue development are likely the primary factors contributing to the increased mortality observed in SCT cubs. Currently, available data have not fully clarified the causal relationship between inbreeding depression and the function of genes involved in lung tissue development. Based on existing evidence, we propose the following research strategies: (1) validate the correlation between lung development-related mutations and inbreeding coefficients through lineage retrospective analysis; (2) evaluate the functional impacts of key mutations on the function of alveolar type II epithelial cells via in vitro cellular experiments.

For strategic population management, we propose developing a scientifically guided breeding-pairing program that prioritizes individuals with low mutation loads as the core breeding population. Furthermore, comprehensive genome-wide screening of neonatally deceased cubs and adult tigers should be conducted to identify key genes or gene combinations associated with mortality. This effort will facilitate the establishment of a SNP microarray-based breeding decision system. These measures are expected to provide a robust scientific foundation for the conservation of the endangered South China tiger.

## 5. Conclusions

This study conducted a comprehensive and systematic analysis of the pathological features of lung tissues from SCT and AT cubs. The results firmly established that hypoxia was a crucial factor contributing to the mortality of cubs within these tiger subspecies. Statistical analysis of morphological parameters revealed that the alveolar walls in SCT cubs were significantly thicker than those in AT cubs. These conditions presumably triggered severe hypoxic shock in the fetus, leading to hypoxic distress and subsequent intrauterine death.

## Figures and Tables

**Figure 1 biology-14-00833-f001:**
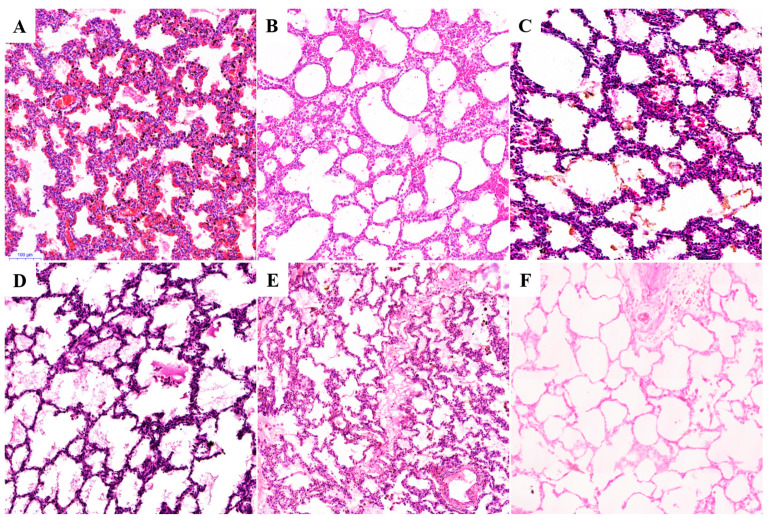
Histology of lung tissue from SCT and AT cubs. (**A**) SCT_stb1. The interstitial vessels were dilated and congested, and abundant brown granular material within the alveolar epithelial cells. (**B**) SCT_stb2. The lung alveoli exhibit fusion and expansion, accompanied by punctate pulmonary contusions. (**C**) SCT_alv. Alveolar spaces were observed to be filled with proteinaceous exudate, whereas localized alveolar lumens contained meconium. (**D**) AT_utd. Filamentous reticulated proteinaceous exudates within the alveolar cavities. (**E**) AT_stb1. A small amount of exudate was present in the alveolar lumen, accompanied by visibly detached epithelial cells. (**F**) AT_stb2. A prominent alveolar fusion expansion phenomenon is observed. The same scale is for all individuals; refer to the bottom left corner of Figure 1A (magnification: 200×).

**Figure 2 biology-14-00833-f002:**
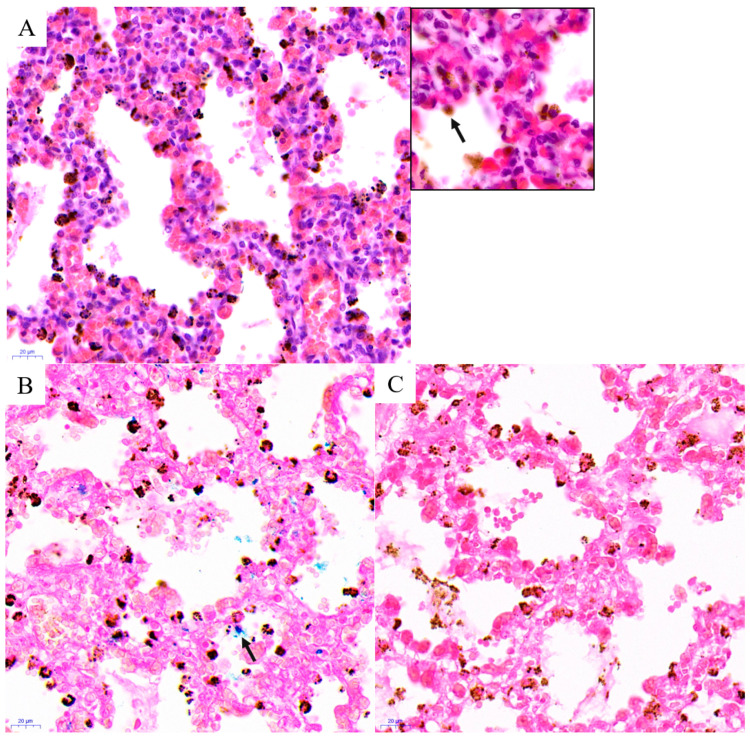
Specific staining of lung tissue in SCT_stb1. (**A**) H&E-stained lung tissue of SCT_stb1. Abundant brown granules inside the alveolar epithelial cells, as indicated by the small box and black arrow (magnification: 1000×). (**B**) Prussian blue stained in individual SCT_stb1. The black arrow denotes the presence of hemosiderin deposits. (**C**) Negative control for Prussian blue staining. A negative control for the Prussian blue stain was conducted using an oxalic acid solution.

**Figure 3 biology-14-00833-f003:**
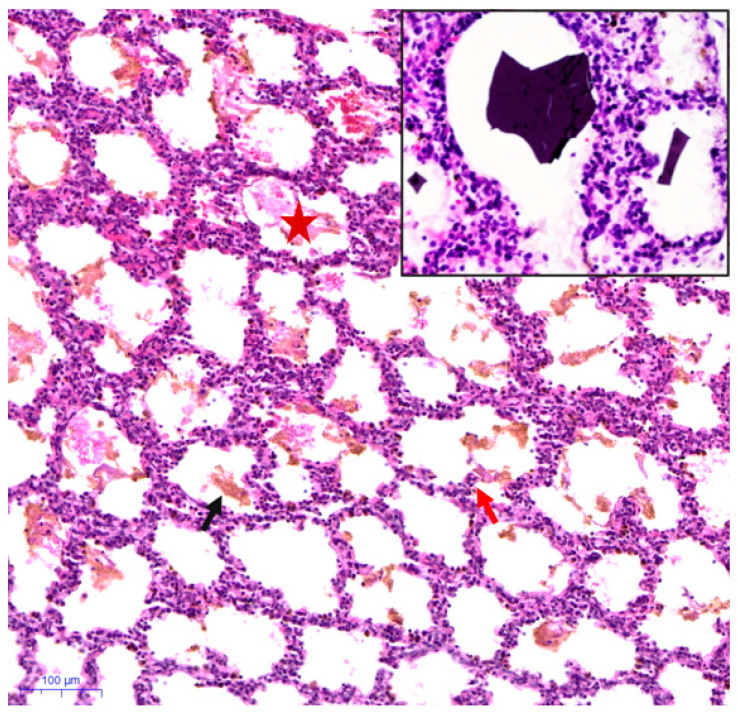
H&E-stained lung tissue of SCT_alv. The black arrow, orange arrow, and red star denote the components of meconium particles, a limited number of epithelial cells, and purplish exudates detected in the alveolar lumen, respectively. Additionally, small boxes show the unusual polygonal or rectangular homogeneous blue crystal-like material observed within the alveolar interstitial space (magnification: 1000×).

**Figure 4 biology-14-00833-f004:**
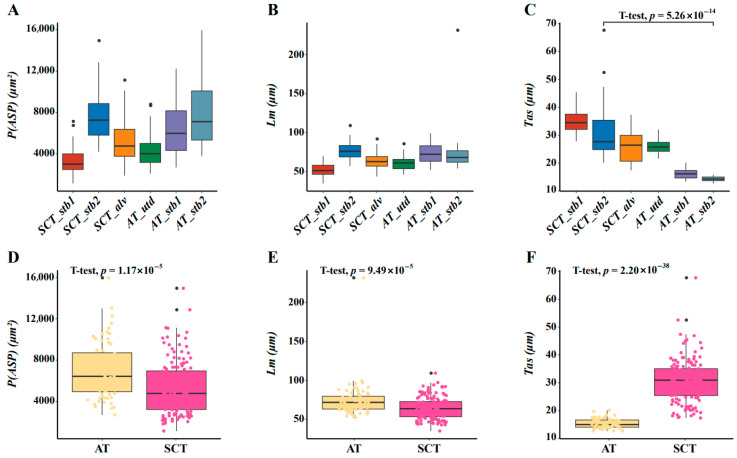
Morphometric analysis of lung histology in dead tiger cubs. (**A**) The degree of alveolar lumen expansion in the six samples. The horizontal axis represents each individual, and the vertical axis corresponds to the area of the alveolar lumen cavity. (**B**) The mean linear intercept of the alveoli in the six samples. The horizontal coordinates refer to Figure 4A, and the vertical coordinate is the magnitude of the mean linear intercepts. (**C**) Thickness of the pulmonary septa in the six samples. The horizontal coordinates refer to Figure 4A, and the vertical coordinate is the value of lung interval thickness. (**D**) The degree of alveolar lumen expansion in the two subspecies of tigers during the same period. The horizontal coordinate AT includes data from AT_stb1 and AT_stb2, and SCT includes data from SCT_stb1, SCT_stb2, and SCT_alv, whereas the vertical coordinate is the area of the alveolar cavity. (**E**) The mean linear intercepts in two subspecies of tigers during the same period. The horizontal coordinates refer to Figure 4D, and the vertical coordinate is the magnitude of the mean linear intercept. (**F**) Lung septal thickness in two subspecies of tigers during the same period. The horizontal coordinates refer to Figure 4D, and the vertical coordinate is the value of the lung interval thickness.

**Table 1 biology-14-00833-t001:** Detailed information on the samples.

Sample ID	Location	Gender	Description of Stillbirth
South China Tiger (*P. t. amoyensis*)			
SCT_stb1	Shanghai Zoo, China	Female	Born dead
SCT_stb2	Luoyang Zoo, China	Female	Died on the day of their birth
SCT_alv	Shanghai Zoo, China	Male	Died on the day of their birth
Amur Tiger (*P. t. altaica*)			
AT_utd	Hengdaohezi Feline Center, China	Male	An intrauterine dead fetus
AT_stb1	Hengdaohezi Feline Center, China	Male	Born dead
AT_stb2	Hengdaohezi Feline Center, China	Female	Died on the day of their birth

**Table 2 biology-14-00833-t002:** Statistics on three parameters of dead tiger cubs.

Sample ID	*P(ASP)* (μm^2^)	*L_m_* (μm)	*Tas* (μm)
SCT_stb1	3360.87 ± 1277.71	53.12 ± 8.10	35.04 ± 3.82
SCT_stb2	7508.86 ± 2338.97	77.07 ± 10.90	31.30 ± 9.75
SCT_alv	5158.74 ± 2135.87	64.43 ± 11.20	25.56 ± 5.50
AT_utd	4305.83 ± 1500.54	61.06 ± 8.33	25.95 ± 2.35
AT_stb1	6340.59 ± 2393.87	74.03 ±12.73	16.17 ± 1.67
AT_stb2	7844.09 ± 2889.88	74.99 ± 31.50	14.27 ± 0.92

Note: The values in this table represent the mean ± standard deviation (SD).

## Data Availability

Lung tissue sections from all individuals are available from the corresponding author upon a reasonable request.

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
