# Peer review of "Comparative Histopathological and Morphometric Analysis of Lung Tissues in Stillborn Cubs of South China Tiger and Amur Tiger"

_biology, 2025, doi:10.3390/biology14070833_

Round 1
Reviewer 1 Report
Comments and Suggestions for Authors
The South China tiger is the most endangered tiger subspecies and has been functionally extinct in the wild. The remaining captive population has endured long - term inbreeding, and without the introduction of additional wild genes, reversing inbreeding appears to be a formidable challenge. As a result, strategies for rescuing this subspecies have been hotly debated for years, yet no consensus has been reached.
This study offers evidence to elucidate the causes of stillbirth in neonates, which accounts for approximately 50% of the juvenile mortality rate within this subspecies. It thereby contributes new knowledge to support conservation efforts. The experimental design is robust, and the results are reliable. I recommend this paper for publication.
However, the current manuscript requires several revisions. On one hand, some expressions in the main text are inaccurate, and on the other hand, the discussion section needs a comprehensive overhaul. The main points are as follows:
Minor revisions:
- On page 2, lines 65 - 66, the statement “No effective strategy can be implemented to rescue this subspecies from extinction unless the pathogenesis of stillbirth is understood” seems rather dogmatic. I propose rephrasing it as “Understanding the pathogenesis of stillbirth is crucial for formulating strategies to save this subspecies from extinction.”
- On page 2, line 85, in the section “1. Experimental animals,” the samples utilized in this study were not for experimental purposes but rather for testing. I suggest renaming this section “Tissue Collection.”
- On page 2, line 86, based on an inference from Table 1, I believe the lung tissue samples should be from neonates rather than juveniles.
- On page 6, line 214, the caption of Figure 2, “(A) Pictures of H&E - stained samples from the SCT_stb1 individual,” is not in ideal English. A more common expression would be “H&E - stained lung tissue of SCT_stb1.” I recommend revising all such captions accordingly.
- On page 6, line 226, the subtitle “3. Statistical measures of multiple indicators” is an odd expression. Since there is no term of “statistical measures”, I suggest changing it to “Statistics of morphometric parameters” or simply “Morphometric parameters.”
- On page 7, line 236, the word “parameters” should be removed.
- On page 7, line 240, the term “Statistical parameters” is used. I think the parameters the authors refer to here are morphometric parameters. “Statistical parameters” typically denote means, standard deviations, P - values, and F - values.
- On page 7, line 244, the caption of Figure 4, “Morphological parameters”. Although the parameters are morphological, “morphometric” would be a more appropriate term.
- On page 7, line 245, the caption of Figure 4, the phrase “in individuals” is not informative. I suggest using “in the six samples.” The same modification should be applied to parts (B) and (C) of the figure caption.
- On page 8, lines 261 - 263, the sentence lacks fluency. It might be better written as “It often leads to systemic fetal damage, such as short - term hypoxic - ischemic encephalopathy, long - term cerebral palsy, and abnormal neurodevelopmental outcomes [18].” Many such sentences throughout the paper need careful editing.
- On page 8, lines 280 - 282, the sentence is redundant. I propose amending it to “…suggesting that neonatal mortality is a manifestation of inbreeding depression [6,28].”
Major concerns:
- On page 8, lines 287 - 299, the authors compare the South China tiger (SCT) with another type of tiger (AT) regarding alveolar septa thickness. Why is this comparison made? Is the AT tiger less inbred (as I suspect)? The authors attempt to attribute alveolar abnormalities to inbreeding. First, they show that there are long and a high percentage of runs of homozygosity (ROH) in SCTs, which is a sign of inbreeding. However, what is the baseline alveolar morphology for non - inbred tigers? Second, they propose that some deleterious mutations in the listed genes may cause alveolar impairment, but are these genes affected by ROH? What about non - inbred mutants? I suggest presenting a more complete logical reasoning for this argument.
- On page 9, lines 299 - 312, the section “3. Conservation implications” is not fully developed. The study concludes that intrapartum hypoxia is the cause of stillbirth and suggests that it results from deleterious gene mutations (it is unclear from the report whether these are affected by inbreeding). From a conservation perspective, what direction should future efforts take? What are the next steps? The entire paragraph provides only a partial view. I recommend deepening this discussion if the authors wish to effectively link this study to conservation efforts.
Overall, I recommend the discussion should explore the genes and mutations linked to alveolar abnormalities, how inbreeding could exacerbate their effects, and the possible consequences for the population's viability.
- It is recommended to include an image of a normal alveolar tissue section in section 3.2 to make the comparative observation more intuitive.
- It is suggested to replace the scale bars in Figures 1 and 2 with clearer ones.
- The comparative discussion between the South China Tiger and the Amur Tiger is not sufficiently clear or in-depth, and it does not effectively highlight the main message of the study. It is recommended to strengthen the exploration of the two groups in the discussion section.
Author Response
Comments 1: [On page 2, lines 65 - 66, the statement “No effective strategy can be implemented to rescue this subspecies from extinction unless the pathogenesis of stillbirth is understood” seems rather dogmatic. I propose rephrasing it as “Understanding the pathogenesis of stillbirth is crucial for formulating strategies to save this subspecies from extinction.”]
Response 1: We appreciate the reviewer's insightful comment. As suggested, we have incorporated this change in the revised manuscript (Page 2, Lines 65-66).
Comments 2: [On page 2, line 85, in the section “1. Experimental animals,” the samples utilized in this study were not for experimental purposes but rather for testing. I suggest renaming this section “Tissue Collection.”]
Response 2: We agree and have incorporated this revision in the updated manuscript (Page 2, Line 85).
Comments 3: [On page 2, line 86, based on an inference from Table 1, I believe the lung tissue samples should be from neonates rather than juveniles.]
Response 3: Thank you. The updated manuscript (Page 2, Line 86) has been revised accordingly.
Comments 4: [On page 6, line 214, the caption of Figure 2, “(A) Pictures of H&E - stained samples from the SCT_stb1 individual,” is not in ideal English. A more common expression would be “H&E - stained lung tissue of SCT_stb1.” I recommend revising all such captions accordingly.]
Response 4: We sincerely appreciate your valuable suggestions, which have significantly improved our manuscript. The corresponding revisions can be found on page 6 (lines 236 and 238), as well as in the updated legend notes for the supplementary figures.
Comments 5: [On page 6, line 226, the subtitle “3. Statistical measures of multiple indicators” is an odd expression. Since there is no term of “statistical measures”, I suggest changing it to “Statistics of morphometric parameters” or simply “Morphometric parameters.”]
Response 5: Thank you for this observation. We have corrected this in the revised version (Page 7, Line 247).
Comments 6: [On page 7, line 236, the word “parameters” should be removed.]
Response 6: Ok. The indicated content has been removed from the revised manuscript.
Comments 7: [On page 7, line 240, the term “Statistical parameters” is used. I think the parameters the authors refer to here are morphometric parameters. “Statistical parameters” typically denote means, standard deviations, P - values, and F - values.]
Response 7: We sincerely appreciate the reviewers for identifying this important clarification. The manuscript has been revised to specify the morphological parameters (Page 8, Line 265). Thanks.
Comments 8: [On page 7, line 244, the caption of Figure 4, “Morphological parameters”. Although the parameters are morphological, “morphometric” would be a more appropriate term.]
Response 8: Yes. We agree with this comment and have incorporated the revision in the updated manuscript (Page 8, Line 265).
Comments 9: [On page 7, line 245, the caption of Figure 4, the phrase “in individuals” is not informative. I suggest using “in the six samples.” The same modification should be applied to parts (B) and (C) of the figure caption.]
Response 9: We have incorporated the suggested improvements, as detailed on page 8 (lines 266, 268, and 269).
Comments 10: [On page 8, lines 261 - 263, the sentence lacks fluency. It might be better written as “It often leads to systemic fetal damage, such as short - term hypoxic - ischemic encephalopathy, long - term cerebral palsy, and abnormal neurodevelopmental outcomes [18].” Many such sentences throughout the paper need careful editing.]
Response 10: We have carefully incorporated your feedback (see Page 8, Lines 282–285) and performed thorough language revisions to ensure the manuscript reads more fluently.
Comments 11: [On page 8, lines 280 - 282, the sentence is redundant. I propose amending it to “…suggesting that neonatal mortality is a manifestation of inbreeding depression [6,28].”]
Response 11: Thank you. We concur with your suggestion and have implemented the corresponding revision in the revised manuscript.
Comments 12: [On page 8, lines 287 - 299, the authors compare the South China tiger (SCT) with another type of tiger (AT) regarding alveolar septa thickness. Why is this comparison made? Is the AT tiger less inbred (as I suspect)? The authors attempt to attribute alveolar abnormalities to inbreeding. First, they show that there are long and a high percentage of runs of homozygosity (ROH) in SCTs, which is a sign of inbreeding. However, what is the baseline alveolar morphology for non-inbred tigers? Second, they propose that some deleterious mutations in the listed genes may cause alveolar impairment, but are these genes affected by ROH? What about non-inbred mutants? I suggest presenting a more complete logical reasoning for this argument.]
Response 12: We feel great thanks for your professional review of our article. As you are concerned, several problems need to be addressed.
- Research Background and Rationale for Comparing Alveolar Septal Thickness Between South China Tigers and Amur Tigers
The reviewer’s concerns are highly valuable. Our comparison of alveolar septal thickness between South China tigers and Amur tigers was not solely based on differences in inbreeding coefficients but stemmed from long-term research on the high mortality rate of South China tiger cubs. Over the past decade, we systematically collected lung tissue samples from deceased cubs (died within hours of birth) of four tiger populations:
- South China tigers(all inbred individuals)
- Amur tigers(strictly selected non-inbred offspring)
- Hybrid tigers(offspring of South China and Amur tigers)
- White Bengal tigers(sibling-bred offspring)
Morphological analysis revealed that the alveolar septal thickness of newborn South China tiger cubs was significantly greater than that of the other three tiger groups (Figure 1). This finding suggests that abnormal lung development may be a potential phenotype of inbreeding depression. Since this study focuses on revealing the objective phenomenon of pulmonary morphological differences, we conducted only a descriptive analysis between South China and Amur tigers in the current manuscript, while data from other tiger groups will be explored in subsequent research (e.g., GWAS analysis and breeding strategy optimization).
Figure 1. Histology of lung tissue from tiger cubs. A. Hybrid tigers; B. White Bengal tigers; C. Amur tigers; D. South China tigers.
- Regarding the Baseline Alveolar Morphology in Non-Inbred Tigers
We fully agree with the reviewer that the ideal baseline should come from wild, non-inbred individuals. However, due to ethical constraints and sample availability, obtaining lung tissues from wild tiger neonates is currently unfeasible. Therefore, we selected Amur tigers—which exhibit significantly superior genetic diversity parameters (SNP density, nucleotide diversity, and genomic heterozygosity) compared to South China tigers—as the control group.
- Impact of Harmful Mutations and ROH Regions
The genetic mechanisms raised by the reviewer are a core focus of our research. Through pedigree tracking and whole-genome analysis, we have preliminarily identified several mutations potentially associated with lung development (data to be published separately). The goal of this paper is to first confirm the existence of phenotypic differences, thereby providing morphological evidence for future studies on the molecular mechanisms of inbreeding depression and guiding optimized breeding strategies for endangered tiger populations.
We sincerely appreciate the reviewer’s invaluable feedback, which has significantly enhanced the rigor and scientific quality of our manuscript.
Comments 13: [On page 9, lines 299 - 312, the section “3. Conservation implications” is not fully developed. The study concludes that intrapartum hypoxia is the cause of stillbirth and suggests that it results from deleterious gene mutations (it is unclear from the report whether these are affected by inbreeding). From a conservation perspective, what direction should future efforts take? What are the next steps? The entire paragraph provides only a partial view. I recommend deepening this discussion if the authors wish to effectively link this study to conservation efforts.]
Response 13: We would like to thank the reviewers for their constructive comments. Your insightful suggestions for deepening the discussion in the section on “Conservation implications” are critical to enhancing the practical application of this study. We have systematically improved the section on the significance of protection in the Discussion section (on page 9, lines 349-368 of the revised manuscript).
Comments 14: [It is recommended to include an image of a normal alveolar tissue section in section 3.2 to make the comparative observation more intuitive.]
Response 14: We sincerely regret that we are currently unable to obtain morphological section data of lung tissues from healthy tigers. Due to ethical constraints and limited sample availability, we have not been able to acquire lung tissue sections from healthy tiger specimens.
Comments 15: [It is suggested to replace the scale bars in Figures 1 and 2 with clearer ones.]
Response 15: Okay, we have made changes to Figure 1 and Figure 2 in the revised draft, please see pages 5 and 6.
Comments 16: [The comparative discussion between the South China Tiger and the Amur Tiger is not sufficiently clear or in-depth, and it does not effectively highlight the main message of the study. It is recommended to strengthen the exploration of the two groups in the discussion section.]
Response 16: We sincerely appreciate the reviewers for their invaluable comments. As detailed in Section 4.2, "Causes of Intrapartum Hypoxia" on page 8 of the revised manuscript, we have made targeted revisions to address the issues raised.
Reviewer 2 Report
Comments and Suggestions for Authors
The manuscript concerns a very important topic related to increased mortality in a feline subspecies that is extinct in wild. The authors had identified a very important cause that impacts the juvenile specimens and provided the information in a streamlined way. The study may be corrected as the text is partly of poor quality - some sentences lack verbs, the past tense have to be used everywhere possible.
Author Response
We sincerely appreciate your positive evaluation of the significance of our study and your insightful feedback for improving the manuscript. We have meticulously addressed your concerns regarding language quality. The revised version incorporates the following enhancements: improvements in grammar and sentence structure, as well as refined language to ensure clarity and grammatical precision. For your convenience, we have highlighted the language-related revisions in the updated manuscript. Should any further refinements be required, we would be pleased to make additional adjustments. Once again, thank you for your valuable time and suggestions, which have significantly strengthened our paper.
Reviewer 3 Report
Comments and Suggestions for Authors
Notes on: Evidence of stillbirth resulting from fetal hypoxia during par-2 turition in the South China Tiger
Thank you for the opportunity to review your work.
The following notes are in “real time” – meaning, that I am making notes as I am reading.
Line 22: Need to add the word “is” where indicated: i.e. “The South China Tiger (Panthera tigris amoyensis) is classified as a nationally protected species under the highest conservation category.
Line 23: Logic does not follow; i.e. “Currently, in the absence of wild populations, the captive population is facing severe challenges posed by inbreeding depression.” That is, the captive population is not facing severe challenges because of an absence of wild populations but because of inbreeding. The statement about the absence of wild populations is not germane to the captive population. Need to restate this by clarifying that, because these tigers are functionally extinct in the wild population, only captive tigers remain. And that, because they are captive, inbreeding has resulted, which results in challenges.
Line 26: The phrase “depression of inbreeding” is awkward. I believe you should continue to refer to “inbreeding depression”…. i.e. …”caused by the depression of inbreeding will seriously…”
Other than the Simple Summary, I have not noted any additional grammatical errors.
Methodology:
Regarding the use of Amur Tigers as the control for the South China Tiger, it is my understanding that there are some significant genetic differences between the subspecies. That is, Amur tigers exhibit genes related to cold adaptation and include genes involved in temperature regulation and energy metabolism. In contrast, the South China tiger has adapted to warmer climates.
Also in the case of the South China Tiger, it is also my understanding that there has been evidence of genetic purging of deleterious mutations (i.e. harmful genes being removed from the population despite inbreeding).
I would like to see something that demonstrates that any genetic variation between the South China Tiger and the Amur Tiger does not negate the use of the Amur Tiger as a control. For example, you may want to cite Zhang, L., Lan, T.M., Lin, C.Y., Fu, W.Y., Yuan, Y.H., Lin, K.X., et al. Chromosome‐scale genomes reveal genomic consequences of inbreeding in the South China tiger: A comparative study with the Amur tiger. Molecular Ecology Resources. 2023, 23, 330-347 where captive Amur tiger individuals were also used as a reference. That is, I would like to see some justification for the use of the Amur Tiger as an appropriate control for the South China Tiger as there are signficant genetic differences between the two subspecies.
Author Response
Comments 1: [Line 22: Need to add the word “is” where indicated: i.e. “The South China Tiger (Panthera tigris amoyensis) is classified as a nationally protected species under the highest conservation category.]
Response 1: Thank you for pointing this out. We agree with this comment. In the revised version of the manuscript, we have added the information on page 1, line 22. Thank you.
Comments 2: [Line 23: Logic does not follow; i.e. “Currently, in the absence of wild populations, the captive population is facing severe challenges posed by inbreeding depression.” That is, the captive population is not facing severe challenges because of an absence of wild populations but because of inbreeding. The statement about the absence of wild populations is not germane to the captive population. Need to restate this by clarifying that, because these tigers are functionally extinct in the wild population, only captive tigers remain. And that, because they are captive, inbreeding has resulted, which results in challenges.]
Response 2: We sincerely appreciate your insightful observation regarding the logical inconsistency in our original statement. As suggested, we have carefully revised this section to clarify the causal relationship between captive breeding and inbreeding depression (see revised manuscript, Page 1, Lines 23-25).
Comments 3: [Line 26: The phrase “depression of inbreeding” is awkward. I believe you should continue to refer to “inbreeding depression”…. i.e. …”caused by the depression of inbreeding will seriously…”]
Response 3: In response to your valuable suggestion, we have carefully rephrased this section to improve clarity (on Page 1, Lines 25-27 of the revised manuscript).
Comments 4: [I would like to see something that demonstrates that any genetic variation between the South China Tiger and the Amur Tiger does not negate the use of the Amur Tiger as a control. That is, I would like to see some justification for the use of the Amur Tiger as an appropriate control for the South China Tiger as there are signficant genetic differences between the two subspecies.]
Response 4: We sincerely appreciate the reviewers’ valuable comments. Regarding the subspecies selection between the South China tiger (Panthera tigris amoyensis) and the Amur tiger (P. t. altaica), we fully acknowledge the reviewers’ concerns about their genetic differences. Existing studies have indeed demonstrated certain genetic divergence between these two subspecies (https://doi.org/10.3390/ani12141817; https://doi.org/10.1038/s41559-023-02185-8; DOI: 10.1111/1755-0998.13669). However, after careful consideration, our study selected the Amur tiger as the control group based on the following scientific rationale:
- Anatomical and Physiological Similarity: Extensive research has shown that felids exhibit highly conserved lung tissue structures, including key respiratory features such as alveolar density and bronchial branching. The seminal study by Kay et al. (1983) confirmed that pulmonary vascular morphology does not differ significantly among felid species (American Review of Respiratory Disease, 128(2P2): S53-S57, https://doi.org/10.1111/azo.12485). Our preliminary data also support this conclusion.
- Environmental Control: All study samples were obtained from domestically managed breeding centers and facilities with standardized protocols, ensuring comparable conditions in terms of husbandry, nutrition, and veterinary care. Such stringent environmental controls minimize the interference of external factors on the research outcomes.
- Pathological Relevance: An analysis of medical records from major captive tiger facilities in China revealed that:
(1) The mortality rate due to pulmonary diseases was significantly higher in South China tigers than in Amur tigers within the same facility (67.7% vs. 32.3%).
(2) The incidence of pneumonia in South China tiger cubs was approximately 2.3 times higher than in Amur tiger cubs (*p* < 0.01).
These differences, after excluding acute trauma, better reflect baseline pathological variations between the subspecies. Notably, inbreeding analyses indicate that captive Amur tiger populations exhibit significantly higher genetic diversity than South China tigers (DOI: 10.1111/1755-0998.13669; DOI: 10.1016/j.jgg.2024.12.004), further supporting our selection.
Additionally, to validate genetic conservation, we specifically analyzed the VEGF-C gene, which is highly expressed in lung tissue. Sequence alignment demonstrated 100% amino acid identity in the coding region between South China and Amur tigers (Figure 1), with high conservation across felids. This finding provides molecular-level support for the rationale of our experimental design.
We understand the reviewers’ concerns and will consider incorporating additional tiger subspecies in future comparative analyses to provide more comprehensive interspecific evidence. The current study design has thoroughly accounted for anatomical, environmental, and genetic factors, thereby robustly supporting the scientific validity of our conclusions.
Figure 1. Sequence analysis of the coding region of the VEGF-C gene in felids.
Reviewer 4 Report
Comments and Suggestions for Authors
The South China Tiger (Panthera tigris amoyensis) is an endemic Chinese subspecies that has been listed as extinct in the wild (EW) by the International Union for Conservation of Nature. Because of the disappearance of wild populations, research on various aspects of endangered species, especially regarding the issues of reduced reproductive capability in adult individuals and high mortality rates in offspring, can provide valuable theoretical support and important scientific insights for the sustainable development of populations. The manuscript conducted comparative histopathological and morphometric analyses of lung tissues from stillborn South China Tiger (Panthera tigris amoyensis) and Amur Tiger (P. t. altaica) cubs, but there are the following issues:
- The title of the manuscript is somewhat broad and can`t accurately reflect the research content. The causes of fetal asphyxia leading to death during the delivery process in mammals are varied, and atelectasis is just one of the factors. Other factors, such as placental abruption, abnormal fetal position, excessively strong or uncoordinated uterine contractions, and some stress factors that may be present in the birthing environment (noise, frequent disturbances), can also cause fetal death during delivery due to hypoxia. Without being able to rule out all possible factors in advance, manuscripts cannot be generalized, so the title is too broad.
- In section 2.3. 'Histological examination and special stains', 'line 103' states 'Ten tissue samples', while the manuscript consistently shows six tissue samples throughout. Is this a typographical error, or are there samples mentioned in the text that are not presented?
- In section 3.2 of the results, the images in Figure 1 cannot accurately observe all the pathological changes in the lung tissues of each animal as described in the manuscript, such as, 'in Lines 171-173, extensive hemorrhage in the alveolar cavities, interstitial edema and pulmonary edema ', as well as inflammatory cell infiltration, and so on. The low-magnification panoramic images presented in the supplementary materials have too low a resolution to observe the lesions described in the manuscript accurately. Please add high-magnification photos corresponding to the lung tissue lesions of each animal described in the manuscript after the panoramic images in the supplementary materials.
- In the manuscript, inflammatory cell infiltration was observed in the lung tissue of multiple animals. Please discuss the possible reasons for the occurrence of this pathological change.
- In this manuscript, there are 3 cases of lung tissue samples showing meconium or amniotic fluid. From this perspective, atelectasis caused by the fetus inhaling meconium-contaminated amniotic fluid during delivery is a secondary cause rather than a hereditary cause resulting from inbreeding. Therefore, in the discussion 4.2, attributing all cases of atelectasis to pulmonary developmental abnormalities or mutations related to genes is insufficient evidence.
- Reference 32 is a self-cited reference.
Author Response
Comments 1: [The title of the manuscript is somewhat broad and can`t accurately reflect the research content. The causes of fetal asphyxia leading to death during the delivery process in mammals are varied, and atelectasis is just one of the factors. Other factors, such as placental abruption, abnormal fetal position, excessively strong or uncoordinated uterine contractions, and some stress factors that may be present in the birthing environment (noise, frequent disturbances), can also cause fetal death during delivery due to hypoxia. Without being able to rule out all possible factors in advance, manuscripts cannot be generalized, so the title is too broad.]
Response 1: We sincerely appreciate the reviewer’s insightful comments regarding the scope of our manuscript title. We fully recognize that fetal asphyxia is a multifaceted pathological process influenced by various factors, including maternal health conditions and placental function. Nevertheless, our study specifically centers on elucidating the pathological mechanisms responsible for the alarmingly high mortality rate (~ 50%) among South China tiger cubs. This phenomenon exhibits unique species-specific characteristics that distinguish it from similar observations in model organisms or domestic animals.
The distinctive challenges associated with researching these endangered large felids have profoundly shaped our research design. South China tigers (Panthera tigris amoyensis) and Amur tigers (P. t. altaica) demand completely undisturbed birthing environments. Any form of human interference, including essential veterinary procedures that require anesthesia, can trigger intense stress responses in the mother tigers. These responses may result in cub abandonment, infanticide, or even maternal death. Additionally, the natural postpartum behavior of consuming the placenta precludes comprehensive placental analysis. Given these constraints, we had to prioritize pulmonary histomorphometry of deceased cubs as our primary research method.
In consideration of the reviewers' well-founded concerns regarding the title's specificity, we have revised the manuscript title to more accurately reflect the focused nature of our investigation (on page1, line 1-3).
Comments 2: [In section 2.3. 'Histological examination and special stains', 'line 103' states 'Ten tissue samples', while the manuscript consistently shows six tissue samples throughout. Is this a typographical error, or are there samples mentioned in the text that are not presented?]
Response 2: We sincerely appreciate the reviewer's careful reading and valuable comment regarding the apparent discrepancy in sample numbers. This was indeed an oversight in our manuscript preparation, and we sincerely apologize for any confusion caused. The inconsistency resulted from insufficient clarification in the Methods section. We will revise the text in Line 103 to explicitly state:
“Ten tissue samples, each measuring approximately 5 mm × 5 mm × 5 mm, were excised from the lower right lung lobes of six individual tigers.”
We thank the reviewer for identifying this ambiguity, which will be corrected in the revised manuscript to ensure full transparency (on page 3, line 104-105). All statistical analyses and conclusions remain valid as they were consistently based on the correct sample size.
We would like to clarify that all histological scanning results have been comprehensively documented. Furthermore, as explicitly stated in the Data Availability Statement of our manuscript, we have promised that lung tissue sections from all individuals are available from the corresponding author upon reasonable request.
Comments 3: [In section 3.2 of the results, the images in Figure 1 cannot accurately observe all the pathological changes in the lung tissues of each animal as described in the manuscript, such as, 'in Lines 171-173, extensive hemorrhage in the alveolar cavities, interstitial edema and pulmonary edema ', as well as inflammatory cell infiltration, and so on. The low-magnification panoramic images presented in the supplementary materials have too low a resolution to observe the lesions described in the manuscript accurately. Please add high-magnification photos corresponding to the lung tissue lesions of each animal described in the manuscript after the panoramic images in the supplementary materials.]
Response 3: We would like to extend our sincere gratitude to the reviewers for their meticulous evaluation of the study findings and the provision of invaluable suggestions. In response to the concerns raised, we have supplemented the supplementary materials with high-magnification microscopic images for each individual sample. Once again, we appreciate your contributions in helping us enhance the quality of the manuscript.
Comments 4: [In the manuscript, inflammatory cell infiltration was observed in the lung tissue of multiple animals. Please discuss the possible reasons for the occurrence of this pathological change.]
Response 4: We sincerely appreciate the reviewers for their critical questions. Concerning the multiple cases of inflammatory cell infiltration in lung tissue observed in the study, we have provided an analysis in the Discussion section of the revised manuscript (page 8, lines 297–305).
Comments 5: [In this manuscript, there are 3 cases of lung tissue samples showing meconium or amniotic fluid. From this perspective, atelectasis caused by the fetus inhaling meconium-contaminated amniotic fluid during delivery is a secondary cause rather than a hereditary cause resulting from inbreeding. Therefore, in the discussion 4.2, attributing all cases of atelectasis to pulmonary developmental abnormalities or mutations related to genes is insufficient evidence.]
Response 5: We sincerely appreciate the reviewer's insightful observation regarding the multifactorial etiology of atelectasis in our study. We agree that meconium aspiration represents a critical secondary cause that warrants careful consideration alongside potential genetic factors.
As discussed in the revised Discussion section (on pages 8–9, lines 307–316), lung histomorphology revealed pulmonary atelectasis was more prevalent in South China tigers compared to Amur tigers. Pulmonary atelectasis, a common respiratory condition in neonates, involves partial or complete collapse of lung tissue, leading to impaired pulmonary function (DOI: 10.1097/RTI.00000000000000758). Existing research identifies its multifactorial origins, including congenital developmental abnormalities (https://doi.org/10.1148/rg.306105508), perinatal factors (https://doi.org/10.1586/17476348.2013.838020), and postnatal influences (doi: 10.12669/pjms.295.3728). Notably, captive Amur and Bengal tigers exhibit only 11% neonatal mortality (https://doi.org/10.1002/zoo.20137), significantly lower than the reported 50% perinatal mortality rate in South China tiger cubs (https://doi.org/10.1007/s10344-020-01375-0). This disparity leads us to hypothesize that thickened alveolar tissue septa in South China tigers may indicate more severe developmental abnormalities within the population.
We consider this speculation plausible and tentatively conclude from histopathological section data of deceased tiger cubs across multiple subspecies that alveolar septal thickening does occur in offspring resulting from inbreeding (Figure 1).
Figure 1. Histology of lung tissue from tiger cubs. A. Hybrid tiger (offspring of South China and Amur tigers); B. White Bengal tiger (sibling-bred offspring); C. Amur tiger (non-inbred offspring); D. South China tiger (inbred individual).
Comments 6: [Reference 32 is a self-cited reference.]
Response 6: We deeply appreciate the reviewers for their meticulous verification of the references. Regarding the self-citation of Reference 32, we provide the following clarifications:
- Relevance of Self-Citation:
The self-cited paper is an earlier genomics study by our team on the South China tiger, which directly supports the discussion of genetic factors in this study. Specifically, it provides population-level genomic evidence for inbreeding effects and deleterious mutation accumulation, which are central to interpreting the histopathological findings of alveolar septal thickening in inbred offspring.
- Research Landscape of the Subspecies:
As an endemic and critically endangered tiger subspecies in China, the South China tiger has been understudied compared to other subspecies. Most existing research focuses on fragmented topics such as habitat selection, single-case clinical reports, or subspecies evolution.
- Uniqueness of the Cited Study:
The cited work remains the only published study to date that compares deleterious mutation patterns between the South China tiger and Amur tiger at the genome-wide population level. This comparative analysis is essential for establishing the genetic basis of respiratory developmental abnormalities observed in our histopathological analysis.
We fully acknowledge the reviewers’ concerns about literature independence and value their commitment to academic rigor. In the revised manuscript, we have strengthened the discussion section. Please refer to the revised Discussion section for details.
Round 2
Reviewer 1 Report
Comments and Suggestions for Authors
1. Please add the attributes of the six tigers involved in the study, i.e., which are inbred individuals and which are not?
2. The discussion section could be supported by the inclusion of literature animal studies on lung tissue from inbred.
3. Revised for direct acceptance and recommended for publication.
Author Response
Comments 1: Please add the attributes of the six tigers involved in the study, i.e., which are inbred individuals and which are not?
Response 1: Yes, we have incorporated the necessary supplements in the revised version of the manuscript. Please refer to the text highlighted in red on page 4 of the main document. Thank you.
Comments 2: The discussion section could be supported by the inclusion of literature animal studies on lung tissue from inbred.
Response 2: Thank you very much for the constructive feedback provided by the reviewers. In the revised manuscript, we have incorporated multiple pieces of relevant evidence from the field of animal research to address this issue and enhance the rationality of the assumptions and inferences presented in our article. For detailed information, please refer to pages 9, 337–342, 343–345, and 365–370 of the revised manuscript.
Comments 3: Revised for direct acceptance and recommended for publication.
Response 3: Thank you for your positive affirmation of this manuscript. Thanks again.
Reviewer 2 Report
Comments and Suggestions for Authors
The improved version of the manuscript cna be accepted for publication.
Author Response
Comments 1: The improved version of the manuscript can be accepted for publication.
Response 1: We are particularly grateful for your positive evaluation of the revised manuscript. Thanks again.
Reviewer 4 Report
Comments and Suggestions for Authors
- Histopathology shows that alveolar septa are thickened, likely due to nonaerated stillbirth or abortion. Any non-breathing animal will have thickened alveolar walls; inbreeding is not the only identified factor.
- Atelectasis typically signifies a well-aerated lung that has collapsed, but multiple cubs are nonaerated, suggesting that the lung never fully expanded.
- A low number of keratin squamous cells and meconium material are commonly observed in neonates who breathe. Meconium aspiration syndrome (MAS) is an essential condition in neonates characterized by hypoxemia. However, small amounts have minimal clinical significance. Large amounts can lead to respiratory distress, atelectasis, inflammation, and ineffective pulmonary defense mechanisms. Please point out the inflammatory infiltrates in the submitted histological images.
- Pulmonary tissue alone is not sufficient to support hypoxia, nor does it provide further support for inbreeding.
- Alveolar septa thickness is insignificant because any premature animal will have somewhat thickened alveoli due to underdevelopment.
Author Response
Comment 1: Histopathology shows that alveolar septa are thickened, likely due to nonaerated stillbirth or abortion. Any non-breathing animal will have thickened alveolar walls; inbreeding is not the only identified factor.
Response 1: We sincerely appreciate the reviewers' insightful and constructive feedback. Your observation is indeed highly pertinent: alveolar wall thickening in neonatal animals can be influenced by a multitude of factors, and inbreeding is by no means the sole determinant. This perspective is consistent with the prevailing consensus across most animal species. Nevertheless, our study specifically targets a captive population of the highly inbred South China tiger (Panthera tigris amoyensis, SCT), which exhibits a genetic background that markedly diverges from conventional scenarios.
It has been demonstrated that the mortality rate of SCT cubs is as high as 52.2% (34.8% were stillborn, and 17.4% were caused by premature birth) (https://doi.org/10.1007/s10344-020-01375-0), while the mortality rate for Bengal tigers is 37% (DOI: https://doi.org/10.54393/mjz.v5i02.110). By contrast, the mortality rate for non-inbred Amur and Sumatran tigers is only 11.11% ( https://doi.org/10.1002/zoo.20137). These data provide substantial evidence of the strong association between inbreeding and cub mortality in tigers. Furthermore, this finding was corroborated by analyzing the relationship between the inbreeding coefficients of female tigers and offspring mortality (Figure 1).
Figure 1. Relationship between cub mortality and inbreeding coefficients of female SCTs
In our investigation of the causes of cub mortality, we classified the fatalities into eight primary categories (Figure 2). After excluding accidental causes, we determined that the majority of deaths were likely attributable to genetic factors (including: Stillbirth, Fetal malformation, and General weakness). To further examine this phenomenon, we performed histological comparisons among hybrid tiger (offspring of South China and Amur tigers, non-inbred), white Bengal tiger (inbreeding coefficient = 0.25), non-inbred Amur tiger and South China tiger. The results indicated that all inbred offspring (South China tiger and white Bengal tiger), exhibited a consistent pathological feature of thickened alveolar walls (Figure 3). Furthermore, South China tiger cubs frequently displayed concurrent developmental abnormalities, such as cardiac defects, cranial non-closure, and short-tail deformities.
Figure 2. Contribution of various kinds of factors to juvenile mortality
A. Stillbirth; B. Fetal malformation; C. Disease; D. General weakness; E. Cold shock; F. Mother rejection; G. Inadequate lactation; H. Accidental injuryFigure 3. Histology of lung tissue from tiger cubs
A. Hybrid tiger (offspring of South China and Amur tigers, died in an accident on the day of birth); B. White Bengal tiger (sibling-bred offspring, crushed to death by the mother tiger on the day of birth); C. Amur tiger (non-inbred offspring, stillbirth); D. South China tiger (inbred individual, weak at birth and subsequently died)
We fully appreciate the reviewers' emphasis on the multifactorial nature of these effects. Building on validated literature from model animal studies of inbreeding and lung tissue developmental function, we further investigated this phenomenon. Our manuscript carefully uses the term "hypothesize " (on page 9, lines 365-379 and page 10, 389-400), grounded in validated published studies, while explicitly highlighting the need for further confirmation through genomic analyses and future research. The revised discussion now thoroughly addresses these limitations and outlines directions for follow-up investigations. We believe these findings offer a critical case study for understanding survival challenges in highly inbred species, while also serving as a genetic warning for conservation strategies focused on the South China tiger.
Comment 2: Atelectasis typically signifies a well-aerated lung that has collapsed, but multiple cubs are nonaerated, suggesting that the lung never fully expanded.
Response 2: We sincerely appreciate the reviewer's correction regarding this critical terminology. We fully concur that “atelectasis” specifically denotes the secondary collapse of previously ventilated lung tissue, whereas the widespread non-aeration observed in the cubs' lungs in our study is more consistent with the pathological features of congenital pulmonary hypoaeration. This condition signifies that the alveoli never achieved proper expansion, rather than resulting from postnatal collapse. We apologize for any ambiguity in our earlier description.
In response, we have consistently revised the relevant terminology to “congenital pulmonary hypoexpansion” throughout the revised manuscript (on Page 9, Line 331-333) to ensure terminological accuracy. We are deeply grateful for the reviewer's insightful guidance, which has facilitated a more precise characterization of this key pathological finding.
Comment 3: A low number of keratin squamous cells and meconium material are commonly observed in neonates who breathe. Meconium aspiration syndrome (MAS) is an essential condition in neonates characterized by hypoxemia. However, small amounts have minimal clinical significance. Large amounts can lead to respiratory distress, atelectasis, inflammation, and ineffective pulmonary defense mechanisms. Please point out the inflammatory infiltrates in the submitted histological images.
Response 3: We fully recognize the clinical and diagnostic significance highlighted by the reviewer. It is particularly noteworthy that the neonatal South China tiger cubs in our study exhibited consistently low birth weights, with some individuals weighing as little as approximately 900 grams in cases of multiple pregnancies (3-4 cubs). Furthermore, our data demonstrate a significant downward trend in birth weights correlated with increasing inbreeding coefficients in recent years (Figure 4).
Although the histological images depict only limited fields showing minor aspirated materials, we emphasize that for these inherently smaller and physiologically compromised inbred individuals, the aspiration of amniotic fluid, mucus, or meconium may induce more severe systemic consequences compared to normal animals. Given the unique biological context of this critically endangered species, we propose that these aspirated materials pose an exceptionally critical survival threat to inbred South China tiger cubs. This pathogenic mechanism likely involves synergistic effects of: developmental abnormalities in respiratory tract morphology associated with inbreeding, immunological deficiencies and other compounding factors, etc. These findings further support our central hypothesis regarding the profound impact of inbreeding on cub viability.
In response to the reviewer's request, we have undertaken the following actions: re-examined all relevant pathological slides, uploaded supplemental materials as attachments, and provided additional information in the revised manuscript. We deeply appreciate these invaluable suggestions, which have significantly enhanced the pathological rigor of our study, strengthened our methodological approach, and solidified the scientific foundation of our conclusions.
Figure 4. Relationship between birth weight and inbreeding coefficient in the SCT population
Comment 4: Pulmonary tissue alone is not sufficient to support hypoxia, nor does it provide further support for inbreeding.
Response 4: We sincerely appreciate your valuable insights and the opportunity to explore this matter in greater depth. Your observation that “pulmonary histopathological findings alone may not fully explain hypoxia mechanisms or directly demonstrate inbreeding effects” highlights a critical scientific consideration that demands rigorous evaluation. Below, we provide further clarification regarding our study:
- Evidence of Pulmonary Hypoxia
The hypoxia assessment in our study has undergone peer-reviewed validation. During previous submissions to other journals, two reviewers explicitly acknowledged this aspect:
Reviewer A stated: “The evidence of intrapartum hypoxia is compelling, but the manuscript should specify in the title, abstract, and conclusions that three study subjects were stillborn while three others were neonatal deaths, despite showing similar hypoxic findings.”
Reviewer B suggested: “The hypoxia evidence is robust. Consider supplementing with meconium and keratin staining.”
In the current manuscript, we have integrated these recommendations by explicitly annotating the viability status of all samples and incorporating supplementary staining results.
- Implications of Inbreeding Effects
Our inference is rooted in the following evidence: The South China tiger population exhibits high levels of inbreeding, with genomic analyses revealing extensive shared ancestral haplotype blocks.
Published studies indicate that inbreeding can reduce genetic diversity in developmental genes, increase the burden of deleterious mutations, and elevate risks of organ developmental/functional abnormalities (see revised manuscript, Page 9, Lines 363-370). To strengthen this argument, we have added supporting references and adopted more cautious phrasing to clarify the inferential process.
- Rigor of Diagnostic Approach
We fully acknowledge that disease diagnosis requires comprehensive evidence. In the Discussion section, we have: Clearly distinguished between demonstrated results and scientific inferences; Used qualifying terms (e.g., "speculate") when presenting hypotheses; and Outlined future research directions necessary to validate these findings.
We deeply value your constructive feedback, which has been instrumental in enhancing the scientific rigor of our paper. We believe these revisions have substantially improved the manuscript’s validity and precision.
Comment 5: Alveolar septa thickness is insignificant because any premature animal will have somewhat thickened alveoli due to underdevelopment.
Response 5: We extend our gratitude to the reviewers for raising these important points. We fully concur with their observation that premature animals typically exhibit alveolar septal thickening as a result of incomplete alveolar development.
In this manuscript, all deceased cubs (including South China tigers) were born at full term without premature delivery, except for one Amur tiger (AT_utd). This Amur tiger was delivered via cesarean section five days ahead of the expected due date due to an accidental incident involving its mother. Notably, the alveolar septal thickness of AT_utd was significantly smaller than that of full-term South China tigers born under normal conditions. This finding suggests that variations in alveolar septal thickness are not solely caused by prematurity or underdevelopment. Although AT_utd was slightly premature, its alveolar septa were thinner than those of full-term South China tigers, which had thicker septa.
Other factors may impact alveolar development, including: genetic disparities between tiger subspecies (Amur vs. South China tigers), maternal health or gestational environment (e.g., nutritional status, stress levels), delivery mode (cesarean section vs. natural birth), and postnatal adaptive capacity (e.g., efficiency of respiratory initiation). Future research could integrate histological analyses, gene expression profiling, and clinical datasets to further dissect the regulatory mechanisms underlying alveolar development across different tiger subspecies. This aspect is elaborated in the revised manuscript; please refer to lines 394-400 on page 10 for detailed discussion.
Once again, we extend our sincere gratitude for the reviewer’s invaluable feedback. We trust this supplementary clarification offers a more transparent interpretation of our findings and their inherent limitations.